# Discovery of 1-Benzhydryl-Piperazine-Based HDAC Inhibitors with Anti-Breast Cancer Activity: Synthesis, Molecular Modeling, In Vitro and In Vivo Biological Evaluation

**DOI:** 10.3390/pharmaceutics14122600

**Published:** 2022-11-25

**Authors:** Dusan Ruzic, Bernhard Ellinger, Nemanja Djokovic, Juan F. Santibanez, Sheraz Gul, Milan Beljkas, Ana Djuric, Arasu Ganesan, Aleksandar Pavic, Tatjana Srdic-Rajic, Milos Petkovic, Katarina Nikolic

**Affiliations:** 1Department of Pharmaceutical Chemistry, Faculty of Pharmacy, University of Belgrade, Vojvode Stepe 450, 11221 Belgrade, Serbia; 2Fraunhofer Institute for Translational Medicine and Pharmacology (ITMP), 22525 Hamburg, Germany; 3Fraunhofer Cluster of Excellence for Immune-Mediated Diseases (CIMD), 22525 Hamburg, Germany; 4Group for Molecular Oncology, Institute for Medical Research, University of Belgrade, Dr. Subotića 4, 11129 Belgrade, Serbia; 5Centro Integrativo de Biología y Química Aplicada, Universidad Bernardo O’Higgins, Santiago 8370993, Chile; 6Department of Experimental Oncology, Institute for Oncology and Radiology of Serbia, Pasterova 14, 11000 Belgrade, Serbia; 7School of Pharmacy, University of East Anglia, Norwich Research Park, Norwich NR4 7TJ, UK; 8Institute of Molecular Genetics and Genetic Engineering, University of Belgrade, Vojvode Stepe 444a, 11000 Belgrade, Serbia; 9Department of Organic Chemistry, Faculty of Pharmacy, University of Belgrade, Vojvode Stepe 450, 11221 Belgrade, Serbia

**Keywords:** drug discovery, 1-benzhydryl piperazine, hydroxamic acid, histone deacetylases, breast cancer, zebrafish xenograft model, anti-metastatic effect

## Abstract

Isoform-selective histone deacetylase (HDAC) inhibition is promoted as a rational strategy to develop safer anti-cancer drugs compared to non-selective HDAC inhibitors. Despite this presumed benefit, considerably more non-selective HDAC inhibitors have undergone clinical trials. In this report, we detail the design and discovery of potent HDAC inhibitors, with 1-benzhydryl piperazine as a surface recognition group, that differ in hydrocarbon linker. In vitro HDAC screening identified two selective HDAC6 inhibitors with nanomolar IC_50_ values, as well as two non-selective nanomolar HDAC inhibitors. Structure-based molecular modeling was employed to study the influence of linker chemistry of synthesized inhibitors on HDAC6 potency. The breast cancer cell lines (MDA-MB-231 and MCF-7) were used to evaluate compound-mediated in vitro anti-cancer, anti-migratory, and anti-invasive activities. Experiments on the zebrafish MDA-MB-231 xenograft model revealed that a novel non-selective HDAC inhibitor with a seven-carbon-atom linker exhibits potent anti-tumor, anti-metastatic, and anti-angiogenic effects when tested at low micromolar concentrations.

## 1. Introduction

Triple-negative breast cancer (TNBC) presents an aggressive subtype of breast cancer with inferior survival outcomes, as it lacks molecular biomarkers such as estrogen receptor (ER), progesterone receptor (PR), and human epidermal growth factor receptor 2 (HER2) [1]. Globally, it is estimated that TNBC accounts for 15–20% of patients with breast cancer diagnosis [2]. At the time of diagnosis, the frequency of distant metastasis is found in approximately 40% of patients with TNBC. Availability of endocrine and anti-HER2 therapies is associated with better survival in ER-positive and HER2-receptor-positive diseases, whereas cytotoxic chemotherapy remains as a predominant treatment option for patients with early-stage and advanced-stage TNBC [3,4]. Owing to its high incidence of metastases, the development of small molecules for such a therapeutically elusive disease remains a remarkably challenging field.

In the era of precision oncology, epigenetic alterations are recognized as significant molecular hallmarks that contribute to breast tumorigenesis. Despite the role of epigenetics in cancer initiation events, the activation of invasion and metastasis is connected with epigenetic abnormalities [5,6]. An imbalance in post-translational modifications of histones, such as histone lysine acetylation and deacetylation, is closely linked to tumor initiation and progression [7]. The reversible nature of histone post-translational acetylation is mediated by two families of enzymes, histone acetyltransferases (HATs) and histone deacetylases (HDACs). HDACs can be involved either in cancer initiation events, such as the induction of apoptosis, differentiation, cell cycle arrest, and mitochondrial stress [8], or in cancer progression events, migration, invasion, and angiogenesis [9].

In the human proteome, there are 11 known HDACs (named HDAC1-11) that are zinc-containing metalloenzymes and 7 NAD^+^-dependent histone deacetylases (known as sirtuins 1–7) [10]. The first resolution of the crystal structure of a histone deacetylase-like protein in complex with its inhibitors [11] provided three key components of the general pharmacophore required for HDAC inhibition, namely, a surface recognition group (CAP group), a hydrocarbon linker (aliphatic or aromatic linker), and the zinc binding group (ZBG), which acts as a warhead (Figure 1A) [11]. Thus far, many HDAC inhibitors (HDACi) have been designed and synthesized [12], with five of them currently used in chemotherapy of hematological malignancies (Figure 1B). Hydroxamic acid is the most widely used ZBG, which is present in marketed drugs such as vorinostat, belinostat, and panobinostat. Others include the natural product romidepsin, which has an alternative ZBG, as it is a prodrug that possesses a disulfide bond that, upon metabolism, releases a zinc binding thiol group and the 2-aminobenzamide inhibitor, tucidinostat (Figure 1B).

For more than a decade, HDAC6 has been studied as a promising target for cancer metastasis amongst the human histone deacetylases (HDACs) [13,14,15,16]. Unlike most of the human HDACs, the HDAC6 isoform is localized primarily in the cytosol, and hydrolyzes terminal N-acetyl-lysine residues of a wide spectrum of the non-histone proteins (e.g., α-tubulin, cortactin, and heat shock protein 90) [17,18]. A unique feature of HDAC6 is the presence of two catalytic domains (CD1 and CD2) that show diverse substrate specificity and kinetics [19,20], a feature that contrasts markedly with the class I HDACs (e.g., HDAC1, HDAC2, HDAC3, and HDAC8 isoforms). The outer rim of the HDAC6 isoform is significantly wider than in class I HDACs, which is one of the prerequisites for inhibitors to target the HDAC6 isoform with higher affinity [21]. Its cellular localization and non-epigenetic roles in cancer progression provide a rationale for therapeutic use of selective HDAC6 inhibitors in preclinical metastasis models [22,23,24]. However, selective HDAC6 inhibitors alone were reported to be inadequate as anti-cancer compounds, and it was suggested that an anti-metastatic drug should be a non-selective HDACi or a selective HDAC6i in combination with other chemotherapeutics [25]. It is noteworthy that the FDA-approved HDACi isoforms (Figure 1B) are pan-active hydroxamic acid derivatives (vorinostat, belinostat, and panobinostat) that show anti-metastatic effects in breast cancer through epigenetic control of gene expression (inhibition of class I HDACs) [26].

The question of whether selective HDAC6 or pan-HDAC inhibitors have better anti-metastatic effects is still disputed. A thorough search of the relevant literature revealed that nine isozymes out of eleven metal-dependent HDACs have been linked to the metastatic events in breast cancer (Appendix A). As there are many ambiguous reports on anti-metastatic HDACi, we were encouraged to design HDAC inhibitors with an anti-metastatic effect by use of a single capping group (1-benzhydryl-piperazine), hydroxamic acid as ZBG, and diverse hydrocarbon linkers. Here, we report the synthesis and in vitro and in vivo biological profiling of novel HDAC inhibitors bearing a 1-benzhydryl piperazine scaffold. This culminated in the identification of a selective HDAC6 inhibitor **9b** and a pan-HDAC inhibitor **8b**, both displaying nanomolar potencies against the HDAC6 isoform.

## 2. Materials and Methods

### 2.1. Chemistry—General Procedures

All chemicals and solvents were purchased from commercial sources, and were used without further purification. ^1^H and ^13^C nuclear magnetic resonance (NMR) spectra were recorded on a Bruker Ascend 400 (400 MHz) spectrometer at room temperature, using TMS as an internal standard. Chemical shifts were reported in ppm (δ). Spin multiplicities were described as *s* (singlet), *d* (doublet), *dd* (double doublet), *t* (triplet), or *m* (multiplet). Coupling constants were reported in hertz (Hz). Mass spectral data were recorded using an Agilent Technologies 6520 Q-TOF spectrometer coupled with an Agilent 1200 HPLC, LTQ Orbitrap XL by electrospray ionization (ESI).

### 2.2. Alkylation of 1-Benzhydryl Piperazine (General Procedure A)

In a round-bottom flask, 1 equivalent of 1-benzhydryl piperazine was dissolved in acetonitrile (5 mL), 1.1 equivalents of corresponding bromo-methyl ester were added, along with 1.7 equivalents of potassium carbonate, at 0 °C. The reaction mixture was heated under reflux overnight. The completion of the reaction was followed by TLC (ethyl acetate:dichloromethane = 1:1). When the reaction was completed, the mixture was filtered through Celite and washed with ethyl acetate. The solvents were removed in vacuo, and the mixture was purified with flash chromatography.

### 2.3. Synthesis of Hydroxamic Acid Derivatives (General Procedure B)

Freshly prepared solution of hydroxylamine in methanol was made by mixing methanol solutions of hydroxylamine hydrochloride (3 equivalents) and potassium hydroxide (6 equivalents). The mixture was rigorously stirred on ice bath for 30 min, during which, potassium chloride precipitate was formed. The precipitate was removed by filtration and freshly prepared NH_2_OH was used for the synthesis of hydroxamic acid derivatives.

To a solution of corresponding methyl-ester **2a**–**9a** (1 equivalent) dissolved in MeOH and cooled at 0 °C, freshly prepared methanol NH_2_OH solution (3 equivalents) was added dropwise. The pH of the solution was adjusted with KOH to be approximately 10. The mixture was left to stand at room temperature for 4–6 h. After the completion of the reaction, methanol was evaporated, saturated solution of NaHCO_3_ was added, and the hydroxamic acid derivative was washed with ethyl acetate. The compound was washed with brine (3 × 5 mL) and dried over anhydrous Na_2_SO_4_. The organic solvent was removed under reduced pressure. Purification by flash column chromatography (dichloromethane:methanol = 95:5) afforded the desired hydroxamic acid derivatives.

### 2.4. Molecular Docking

The synthesized inhibitors were sketched in ChemDraw software v. 7.0.1 and the ionization parameters (pH = 7.4) were calculated in MarvinSketch 6.1.0 (https://chemaxon.com/, accessed on 10 November 2021). The energy minimization of the synthesized ligands was performed in the gas phase in Chem3D Ultra 7.0 software (Hartree-Fock 3-21G method by use of Gaussian 7.0.0). Optimized 3D structures of the inhibitors were used as deprotonated hydroxamates for molecular docking study. The crystal structure of the histone deacetylase 6 (PDB: 5EDU) was downloaded from the PDB website (https://www.rcsb.org, accessed on 10 November 2021). The initial enzyme structure was protonated and tautomeric states of residues under physiological conditions were assigned using PDB2PQR 3.1.0 software accessed through a server (https://server.poissonboltzmann.org/pdb2pqr, accessed on 10 November 2021). Catalytic water molecule in HDAC6 was retained for docking calculations. The structures of the inhibitors were treated as flexible, whereas the enzyme was kept rigid. The docking procedure was carried out in metalloenzyme configuration mode in GOLD software v.5.8.1, with ChemScore calculated as the scoring function. The validity of the docking procedures was examined by inspecting the RMSD value (RMSD < 2 Å), and docking poses were visualized in Discovery Studio software v.17.2.0.

### 2.5. HDAC Enzymes Assay

Human recombinant C-ter-His-FLAG-HDAC1 (Catalog #50051), HDAC-3/NcoR2 (Catalog #50003), N-ter-GST-HDAC-6 (Catalog #50006), and C-terminal His-tag-HDAC8 (Catalog #50008) proteins were purchased from BPS Bioscience (San Diego, CA, USA). Trichostatin A (Sigma-Aldrich, St. Louis, MO, USA), dissolved in DMSO, was used as a positive control, whereas DMSO was used as a negative control. Synthesized compounds were dissolved in DMSO (*v*/*v*) and stored at −20 °C. Inhibition profiles for the synthesized compounds were monitored with bioluminogenic HDAC-Glo™ I/II assay (Promega Corp., Madison, WI, USA) [27]. After single-point screening, the compounds were tested at eight different concentrations, with twofold dilution. Compound solutions were dispensed using the Echo 550^®^ into 384-well assay plates (10 nL/well). This was followed by the addition of HDAC enzyme (5 µL/well) using the multidrop liquid handling system and incubated for 10 min at room temperature. The final addition of the HDAC-Glo assay reagent (10 μL per well) was added to initiate the luciferase reaction. After 10 min incubation at room temperature, the luminescence was read on an EnSpire Microplate Reader. All the experiments were performed in triplicate, and the raw data obtained after screening were analyzed using Prism software. The dose–response curves were generated using a four-parameter logistic fit in eight-point format, yielding the IC_50_ values.

### 2.6. Breast Cancer Cell Cultures

Human breast adenocarcinoma cells (MCF-7, ATCC^®^ HTB-22™) and triple-negative breast cancer cells (MDA-MB-231, ATCC^®^ HTB-26™) were obtained from the American Type Culture Collection (Rockville, MD, USA). Cells were maintained as a monolayer culture in DMEM:Ham’s 12 (1:1) medium (Sigma-Aldrich, St. Louis, MO, USA) supplemented with penicillin (192 U/mL), streptomycin (200 µg/mL), and 10% of heat-inactivated fetal calf serum (FCS). Cells were grown at 37 °C in 5% CO_2_ and humidified air atmosphere, by twice weekly subculture.

#### 2.6.1. Cell Viability (MTT) Assay

Cytotoxic activity of synthesized 1-benzhydryl-piperazine-based HDAC inhibitors was assessed on MDA-MB-231 and MCF-7 cells using an MTT assay [28]. MDA-MB-231 (5 × 10^3^ cells/well) and MCF-7 (5 × 10^3^ cells/well) were treated with synthesized compounds in six different concentrations (100, 50, 25, 12.5, 6.25, and 3.13 μM), and each concentration was added in five replicates. MTT solution (3-(4,5-dimethylthiazol-2-yl)-2,5-diphenyltetrazolium bromide) (Sigma-Aldrich, St. Louis, USA) was added to each well (20 µL) after 48 h. After 4 h of incubation, 100 μL of 10% SDS was added in each well and incubated at 37 °C. On the next day, the absorbance at 570 nm was recorded. The ratio of absorbance (570 nm) values between treated and control cells multiplied by 100 was used to calculate cell survival (%). The concentration of synthesized HDAC inhibitors that reduced cell viability by 50% was defined as the IC_50_ value, and was compared to the vehicle control.

#### 2.6.2. Apoptosis Assay

Examined breast cancer cells (MDA-MB-231) were stained with annexin V–fluorescein isothiocyanate/and 7AAD (BD Pharmingen, San Diego, CA, USA), and the samples were prepared following the specifications given by manufacturer. The population of apoptotic cells was analyzed by flow cytometry using a FACS Calibur cytometer and Cell Quest computer software (Becton Dickinson, Heidelberg, Germany).

#### 2.6.3. Cell Cycle Analysis

MDA-MB-231 cells were treated with tested compounds for specified time courses. After treatment, examined cells were washed in cold PBS and incubated for 30 min in 96% ethanol on ice, centrifuged, and incubated with 80 μL RNase A (200 μg/mL/mL) and 50 μL propidium iodide (50 μg/mL) for 30 min at 37 °C. The cell cycle was analyzed by FACS Calibur E440 (Becton Dickinson) flow cytometer and Cell Quest software. Results are presented as a percentage of cell cycle phases.

#### 2.6.4. Detection of Mitochondrial Membrane Potential

Mitochondria membrane potential was determined by the fluorescent dye JC-1 [29]. After treatment with different concentrations of compounds under standard culture conditions for 48 h, the cells were stained with 40 μL of JC-1 (final concentration 15.4 μM) and incubated under standard conditions for 15 min. The fluorescence intensity of cells was determined by flow cytometry.

#### 2.6.5. Generation and Analysis of Tumor Spheres

MDA-MB-231 cells were seeded at a cell density of 1000 c/w in 100 μL of DMEM containing 10% FCS in a low-attachment NunclonSphera 96-well ultra-low attachment plate (Thermo Scientific Nunc™, Rochester, NY, USA), to form multicellular tumor spheres, for 4 days. The formation and growth of tumor spheres were examined and imaged with an Olympus CKX53 microscope, using a 10× objective. MDA-MB-231 tumor spheres were treated by carefully adding 50 μL of the medium with fresh nutrient medium for control spheres or with compound-supplemented medium (50 μM) for treated spheres for another 72 h. The cytotoxicity of compounds toward the MDA-MB-231 tumor spheres was investigated by MTT assay.

In another series of experiments, MDA-MB-231 cells were seeded in co-culture with investigated compounds (IC_50_ and 50 μM) at a cell density of 2000 c/w in 150 μL of DMEM containing 10% FCS in a low attachment NunclonSphera 96-well ultra-low attachment plate (Thermo Scientific Nunc™) to form multicellular tumor spheroids for 4 days. The formation and growth of tumor spheroids were examined and imaged with an Olympus CKX53 microscope, using 4x objective. The cytotoxicity of compounds toward the MDA-MB-231 tumor spheroids was investigated by MTT assay.

#### 2.6.6. Bicameral Motility and Invasion Assay

To determine MDA-MB-231 cell migration and invasion cell capacities, Corning^®^ Costar^®^ Transwell^®^ cell culture inserts with 8.0 μm pore polycarbonate filters (Sigma Aldrich, CLS3464) and Corning™ BioCoat™ Matrigel™ Invasion Chamber with Corning Matrigel Matrix™ cell culture inserts (Thermo fisher scientific, Waltham, MA, USA) respectively were used. Briefly, 6 × 10^4^ cells in 200 μL were seeded in the upper chamber and the bottom chamber was filled with 800 μL of medium. After 24 h under indicated treatments, the remaining cells in the upper chamber were gently removed with a cotton swab, moistened with medium, and the inserts were carefully washed twice with 37 °C warmed PBS. Then, attached cells at the bottom side of the membrane were fixed by immersing in ice cold methanol for 2 min and stained with 0.1% crystal violet for 15 min at room temperature. Stained cells were photographed and quantified using NIH ImageJ software.

#### 2.6.7. Wound Healing Assay

To determine MCF-7 cell migration properties, 1 × 10^5^ cells were seeded in 24-well plates and cultured until confluent. Then, cell monolayers were scratched using a sterile 200 μL pipette tip, and cell cultures were allowed to grow for 24 h under the indicated treatments. At the end of incubation period, culture medium was removed and wells were washed twice with PBS, then, cells were fixed with ice-cold methanol and stained with 0.1% crystal violet. In addition, cell monolayers were fixed and stained just after scratching to represent zero time migration. Finally, cell migration into the scratch area was documented by inverted light microscopy and quantified using NIH ImageJ software [30].

### 2.7. Animal Study in the Zebrafish (Danio rerio) Model

Embryos of wild type (AB) and transgenic Tg(fli1:EGFP) and Tg(-2.8fabp10a:EGFP) zebrafish (*Danio rerio*) lines were kindly provided by Dr. Ana Cvejić (Wellcome Trust Sanger Institute, Cambridge, UK) and raised to adult stage in a temperature- and light-controlled zebrafish facility at 28 °C and standard 14–10 h light–dark photoperiod. Adult fish were regularly fed with commercial dry food (SDS300 granular food; Special Diet Services, Essex, UK; and TetraMinTM flakes, Tetra, Melle, Germany) twice per day and *Artemia* nauplii daily. All experiments involving zebrafish were performed in compliance with the European directive 2010/63/EU and the ethical guidelines of the Guide for Care and Use of Laboratory Animals of the Institute of Molecular Genetics and Genetic Engineering, University of Belgrade.

#### 2.7.1. Acute and Developmental Toxicity Assessment

Toxicity evaluation of the tested compounds in the zebrafish model was carried out following the general rules of the OECD Guidelines for the Testing of Chemicals (OECD, 2013, Test No. 236) [31], and procedures described in the literature [32]. Briefly, wild type (AB) zebrafish produced by pair-wise mating were collected, washed of debris, and distributed into 24-well plates containing 10 embryos/well and 1 mL of E3 medium (5 mM NaCl, 0.17 mM KCl, 0.33 mM CaCl_2_, and 0.33 mM MgSO_4_·7H_2_O in distilled water), and raised at 28 °C.

To assess acute (lethality), inner organ, and developmental (teratogenicity) toxicity, the embryos at the 6 h post-fertilization (hpf) stage were treated with six different concentrations of each tested compound (100, 50, 25, 12.5, 6.25, and 3.13 µM). Stock solutions of test substances were made in DMSO. Embryo water and DMSO (0.25%) were used as negative and positive controls, respectively. The experiment was performed three times using 20 embryos per concentration. Treated embryos were inspected for 22 toxicological parameters (Appendix A) every day by 120 hpf upon a stereomicroscope (Carl Zeiss™ Stemi 508 doc Stereomicroscope, Jena, Germany). Dead embryos were discarded every 24 h. At 120 hpf, embryos were anesthetized by addition of 0.1% (*w*/*v*) tricaine solution (Sigma-Aldrich, St. Louis, MO, USA), photographed, and killed by freezing at −20 °C for ≥24 h.

#### 2.7.2. Anti-Angiogenic Activity Assessment

The inhibitory activity of **8b** on angiogenesis was examined using embryos of the transgenic zebrafish Tg(fli1:EGFP) line with GFP-labeled endothelial cells, as previously described [32]. Briefly, embryos at the 6–8 hpf stage were exposed to three non-toxic concentrations and incubated at 28 °C by 48 hpf. At 48 hpf, the treated embryos were anesthetized with 0.02% tricaine, imaged under a fluorescence microscope (Olympus BX51, Applied Imaging Corp., San Jose, CA, USA), and analyzed for the development of intersegmental blood vessels (ISVs). The experiment was performed three times using ten embryos per concentration. ISV lengths were measured using ImageJ program, and expressed as mean value with standard deviation. Inhibitory effect of the applied treatment was determined in relation to the control (DMSO-treated) group, arbitrarily set to 100%.

#### 2.7.3. Anti-Cancer Activity Evaluation in Human and Zebrafish Cell-Derived Xenografts (CDX)

##### Cell Line Culture Preparation

The human breast carcinoma MDA-MB 231 cell line was cultured in DMEM:Ham’s 12 (1:1) medium (Sigma-Aldrich, St. Louis, MO, USA) supplemented with 10% FBS (Sigma-Aldrich, St. Louis, MO, USA), 100 µg/mL streptomycin, and 100 U/mL penicillin (Sigma-Aldrich, St. Louis, MO, USA), and grown as a monolayer in humidified atmosphere of 95% air and 5% CO_2_ at 37 °C. Prior to microinjection, the cells were washed once with PBS and trypsinized (0.25% trypsin/0.53 mM EDTA) (Sigma-Aldrich, St. Louis, MO, USA) to obtain a single-cell suspension. After centrifugation at 1200 rpm for 5 min, the cells were resuspended in serum-free DMEM medium and labeled with 2 µM Cell Tracker TM Red CMTPX (Thermo Fisher Scientific) according to the manufacturer’s instructions.

#### 2.7.4. Zebrafish Xenografts and Treatment Efficacy Assessment

The zebrafish xenografts with human MDA-MB-231 cells were established according to the previously described procedure with slight modification [33]. A day before the microinjections, Tg(fli1:EGFP) embryos kept at 28 °C were manually dechorionated. At the 48 hpf stage, 5 nL of MDA-MB 231 cell suspension containing 150 labeled cells was microinjected into the yolk of anesthetized embryos by a pneumatic picopump (PV820, World Precision Instruments, Sarasota, FL, USA). The exact number of cells was confirmed by dispensing the injected volume onto a microscope slide and by visual counting. After injection, embryos were incubated, to recover, for at least for 60 min at 28 °C, dead embryos were removed, while alive embryos were transferred into 24-well plates containing 1 mL of E3 water and 10 embryos per well. The injected xenografts were treated with the three doses of **8b** (3.13, 6.25, and 12.5 µM), and maintained at 31–32 °C by 120 hpf. DMSO (0.25%) was used as a negative control. The survival and development of the xenografted embryos was recorded every day until the end of experiment (3 days post injection = 3 dpi), at which point, anesthetized xenografts were processed by fluorescent microscopy. The tumor size, number of xenografts with disseminated tumor cells in the caudal region, and number of disseminated cells per embryo were determined. The tumor size was determined by the fluorescent images using ImageJ. The experiment was repeated two times using 10 embryos per concentration.

### 2.8. Statistical Analysis

The experimental results are expressed as mean values ± SD. The χ2 test was used to determine the differences in anti-angiogenic phenotypes between the untreated and treated groups. In other experiments, the differences between the untreated and treated groups were evaluated using a one-way ANOVA followed by a comparison of the means by Bonferroni test (*p* = 0.05). SPSS 20 software package (SPSS Inc., Chicago, IL, USA) was used to analyze experimental data, whereas graphical representations of the results were prepared in GraphPad Prism 6 (GraphPad Software, Inc., San Diego, CA, USA).

## 3. Results and Discussion

### 3.1. Design and Synthesis of HDAC Inhibitors

Compounds **2b**–**9b** were synthesized using the routes represented in Figure 2. The main idea in designing novel HDACi was to employ the 1-benzhydryl piperazine as a CAP group. Computational fragment-based screening reported in our previous study identified many interesting CAP groups for the design of HDAC6 inhibitors [34]. One such fragment, 1-benzhydryl piperazine, was employed as a template to calculate the total surface area (SAtot) and McGowan volume (Vx) in the Dragon v. 6.0.7. software [35], and to compare these values with the same descriptors calculated for the CAP groups of ACY-1083 [36], tubastatin A [37], and ricolinostat [38]. These inhibitors exhibited HDAC6 selective inhibitory profiles and diversity in the chemistry of CAP groups. Comparing to SAtot and Vx descriptors, we showed that 1-benzhydryl piperazine exerted higher values for the total surface area and McGowan volume (Figure 2), which are important for the inhibitor’s complementarity with the HDAC6 interaction surface area.

The alkylation of 1-benzhydryl piperazine with bromo-alkyl methyl esters (*n* = 1–7) in refluxing acetonitrile produced the 1-benzhydryl piperazine methyl ester derivatives (Exp. Procedure 2.2 and Appendix B) with aliphatic chains (**2a**−**8a**) and a benzyl-methyl ester derivative (**9a**) at room temperature. Finally, methyl esters (**2a**−**9a**) were treated with freshly prepared solutions of hydroxylamine in methanol, yielding eight hydroxamic acid derivatives (**2b**–**9b**, Figure 2, Exp. Procedure 2.3, Appendix B).

### 3.2. Biology

#### 3.2.1. Analysis of HDAC Inhibitory Profiles

The designed and synthesized **2b** to **9b** were screened using a commercial in vitro biochemical luminescence assay kit with purified HDAC enzymes. This assay technology has been adequately validated using reference HDACi with a wide potency range (valproic acid, sodium butyrate, and Trichostatin A) and a variety of HDAC enzymes [27].

To identify selective HDAC6 inhibitors, a preliminary screening at 5 μM concentration of the synthesized compounds was performed to determine the percentage of HDAC6 inhibition. As presented in Table 1, compounds with 1, 2, 3, and 4 carbon atoms in the aliphatic linker (**2b**, **3b**, **4b**, and **5b**) inhibited the HDAC6 isoform by less than 25% at 5 μM.

When considering that the HDAC6 isoform was sensitive to **6b**, **7b**, **8b**, and **9b** (percent inhibition > 90% at 5 μM), we conducted HDAC selectivity studies to determine their IC_50_ values against the HDAC1, HDAC3, HDAC6, and HDAC8 isoforms. Nanomolar inhibitory activities toward the HDAC6 isoform were observed for the 1-benzhydryl derivatives with linker lengths of *n* = 5, 6, and 7, whereas the phenyl-hydroxamic acid derivative (**9b**) showed the most potent HDAC6 inhibition (IC_50_ = 0.031 μM). It can also be concluded from the HDAC selectivity study that all assayed compounds yielded noticeable HDAC6 selectivity (Appendix A), with **9b** displaying the highest selectivity ratio over the HDAC1 (47.5), HDAC3 (112), and HDAC8 (23) isoforms. This observation is in agreement with HDAC6 selectivity inhibition observed for other phenyl-hydroxamic acid derivatives [39]. Within the group of alkyl-hydroxamic acid derivatives, it appears that optimal linker length should be *n* = 5 to retain nanomolar and selective HDAC6 potency (**6b**, IC_50_ = 0.186 μM), which contrasts with the micromolar IC_50_ values toward nuclear isoforms HDAC1/3/8. An increase in the linker length (*n* = 6 for **7b** and *n* = 7 for **8b**) led to a decrease in the selectivity ratios compared to the compound **6b** (Table 2).

To further rationalize the effects of different linkers on the potency and selectivity profiles of the HDACi, a structure-based molecular modeling study of interactions between HDAC6 and synthesized inhibitors was performed. Compounds **6b**, **7b**, **8b**, and **9b** were initially docked into the crystal structure of the second catalytic domain of the human HDAC6 isoform (PBD: 5EDU), and the dynamics of the interaction of selected inhibitors with HDAC6 was investigated by means of molecular dynamics (MD) simulations (Appendix A).

Root-Mean-Square Deviation (RMSD) analysis of HDAC6 backbone atom movements during 100 ns of MD simulations indicated well-converged systems without any larger conformational changes regarding HDAC6 (Appendix A). Visual inspections of trajectories obtained after MD simulations with RMSD analysis of the ligand atoms revealed a relationship between the nature of the linker and conformational flexibility of ligands when interacting with HDAC6. Namely, the presence of an aromatic linker conferred a significant increase in the conformational stability of the **9b** interaction with HDAC6, compared to the ligands with aliphatic linkers (Appendix A). This noticeable difference in the dynamic behavior of ligands with aliphatic and aromatic linkers could be the reason that the highest HDAC6 inhibitory potency was observed in **9b**, which is in agreement with the study of Porter et al. [40].

The predicted binding modes of all studied inhibitors were in alignment with available X-ray crystal structures, where the linker is sandwiched between F620 and F680 of aromatic crevice of HDAC6, and CAP groups interact with L1 and/or L2 pockets at the mouth of HDAC6 active site (Figure 3, Appendix A) [41]. The results of the MD simulations reveal that increasing the length of aliphatic linkers from five to seven carbon atoms contributed to considerable differences in the binding modes of **6b**, **7b**, and **8b**. In contrast, **6b** retained a similar binding mode as ligand **9b**, preferably interacting with the L2 pocket (Figure 3A), and **7b** interacted preferentially with L1 pocket (Figure 3B). Due to the presence of the longest linker in **8b**, the binding mode of this inhibitor was able to shift between L1 and L2 pockets, which is reflected as a bimodal distribution of distances between centers of masses of the CAP group and L1 or L2 pockets (Figure 3C and Appendix A). This observation is in agreement with the binding conformations observed in the X-ray crystal structure of the HDAC6–ricolinostat complex [40].

Based on the results of our studies, **6b** and **9b** represent the most selective HDAC6 inhibitors, and also have similar predicted binding modes (Figure 3A), implying that a linker length of five carbon atoms (*n* = 5) is optimal for achieving selective interaction with HDAC6 in the series of inhibitors with aliphatic linkers. Taken together, the in silico results provide the atomic resolution of the interactions of novel inhibitors with the HDAC6 isoform. Additionally, considering that our selective HDAC6 inhibitors (**6b** and **9b**) uniquely anchor the 1-benzhydryl piperazine moiety in the L2 pocket, substituting phenyl rings in 1-benzhydryl piperazine appears to be a strategy to target the protein landscape in the L2 loop pocket, and for designing novel selective HDAC6 inhibitors.

#### 3.2.2. Examination of Cytotoxic Effects of Synthesized Compounds

The ability of the designed and synthesized HDACi isoforms (**6b**, **7b**, **8b**, and **9b**) to decrease the viability of studied breast cancer cell lines (MDA-MB-231 and MCF-7) was subsequently investigated. As **6b**, **7b**, **8b**, and **9b** are found to be most potent in terms of HDAC6 inhibition, the selective HDAC6 inhibitor tubastatin A was used as a positive control in cell-based assays. Treatment of breast cancer cells with the HDACi isoforms for 48 h resulted in dose-dependent cytotoxicity, as summarized in Table 3 and Appendix A. All tested compounds reduced the viability of MDA-MB-231 breast cancer cells at lower concentrations, compared to MCF-7 cells. The obtained IC_50_ values were 33.40 μM, 10.55 μM, 5.42 μM, and 38.1 μM for **6b**, **7b**, **8b**, and **9b**, respectively. Compound **8b** significantly decreased the viability of MCF-7 cells (IC_50_ = 39.10 μM), whereas the selective HDAC inhibitors **6b** and **9b** reduced cell viability at higher concentrations (IC_50_ values were 84.05 μM and 99.50 μM, respectively). The inhibitor **7b** did not exhibit cytotoxic effects against MCF-7 cells (IC_50_ > 100 μM). Selective HDAC6 inhibitors **6b**, **9b**, and tubastatin A showed moderate cancer cell cytotoxicity, which is consistent with previous studies on breast cancer cell lines [42]. In order to understand the mechanisms of breast cancer cell death upon exposure to tested compounds, we further studied their influence on apoptosis and cell cycle changes.

#### 3.2.3. 1-Benzhydryl-Piperazine-Based HDAC Inhibitors Induce Translocation of Cell Membrane Phosphatidylserine

Initially, the ability of compounds to induce apoptosis of MDA-MB-231 cells was analyzed after 24, 48, and 72 h treatment at IC_50_ concentration by flow cytometry using Annexin V-FITC/7-AAD staining. The induction of early apoptotic cell death was time-dependent in MDA-MB-231 cells upon treatment with **6b**, **7b**, **8b**, **9b**, and tubastatin A (Figure 4A–C).

Twenty-four-hour treatment with **6b** and **7b** induced a slight increase in programmed cell death (Figure 4A). The occurrence of early apoptosis (approximately 20%) after MDA-MB-231 cell treatment with selective HDAC6 inhibitor **6b** was observed after 48 and 72 h (Figure 4B,C). Following 48 h treatment **6b**, **7b**, **8b**, and **9b** induced similar percentages of early apoptotic cells (Figure 4B). Seventy-two-hour treatment of MDA-MB-231 cells with **7b**, **8b**, and **9b** at IC_50_ concentration led to the strong increase in the early apoptotic cell population (around 45%) (Figure 4C). The percentage of early apoptotic cells increased gradually with increasing treatment duration with the selective HDAC6 inhibitor tubastatin A (Figure 4A–C).

Anti-cancer drugs have been shown to activate the intrinsic apoptotic pathway. One of the earliest changes in this process is the dissipation of mitochondrial membrane potential (*∆ψm*) [43]. The flow cytometry data showed a dramatic loss of mitochondrial membrane potential (*Δψm*) in MDA-MB-231 cells, even during 24 h treatment, with all tested compounds (Figure 4D), indicating the initiation of the mitochondrial apoptotic pathway.

#### 3.2.4. Treatment of MDA-MB-231 Cells with Inhibitors Induces Changes in the Cell Cycle

To account for possible anti-proliferative effects of synthesized HDAC inhibitors, we analyzed changes in the cell cycle distribution at IC_50_ concentrations in a time-dependent manner. Treatment of MDA-MB-231 cells with all compounds resulted in a time-dependent increase in MDA-MB-231 dying cell population, followed by a reduction in cells in the G0/G1 phase of the cell cycle (Figure 5). Treatment of MDA-MB-231 cells with **7b** and **8b** at 24 h resulted in a slight increase in the percentage of cells in the G2/M phase, while treatment with **9b** increased the percentage of cells in the S phase of the cell cycle (Figure 5). Moreover, the treatment of cells with **7b**, **8b**, and **9b** after 48 h increased percentage of cells in S phases (Figure 5). Tubastatin A induced an increase in the MDA-MB-231 dying cell population (sub-G1 phase cells around 30% at 72 h) in a time-dependent manner, followed by a reduction in the number of cells in the G0/G1 phase of the cell cycle (Figure 5).

#### 3.2.5. Analyses of the Efficacy of Compounds in a 3D Cancer Model

Three-dimensional (3D) cell cultures may mimic the natural in vivo setting more representatively than 2D cultures since 3D models are beginning to include the cellular morphologies, phenotypes, and interactions seen during in vivo tumor development [44].

A 3D tumor sphere model of MDA-MB-231 cells was introduced to investigate the effects of synthesized compounds at the equimolar concentration (50 µM) on the viability and growth kinetics of tumor spheres after 72 h, in comparison with tubastatin A. After 4 d, formed spheres treated with **8b** for 72 h revealed tenfold lower sensitivities to **8b** (IC_50_ = 50 µM) (Figure 6B) than in 2D cell culture (Table 1). Compound **6b**, as well as tubastatin A, caused a mild decrease in tumor sphere viability at 50 µM concentration, while compounds **7b** and **9b** did not exhibit significant cytotoxic effects in examined spheres (Figure 6B). Moreover, **8b**-treated spheres decreased in size after 72 h of treatment (Figure 6A).

Next, a test of the ability of compounds to inhibit the tumor sphere formation was performed. MDA-MB-231 cells were co-cultured with the compounds (at IC_50_ and 50 µM concentrations) in sphere culture conditions for 4 d (Figure 6C,D). After 4 d, untreated cells formed spheres that had a tightly packed morphology (Figure 6C). Meanwhile, the incubation of MDA-MB-231 cells with **8b** and tubastatin A (50 μM) for 4 d in sphere culture conditions significantly inhibited both sphere formation and spheres viability (Figure 6C,D). In turn, compounds **6b** and **7b**, at higher concentrations, inhibit sphere formation to some extent (Figure 6C,D).

#### 3.2.6. Compound **8b** Decreases Migration and Invasion of Triple-Negative Breast Cancer MDA-MB-231 Cell Line In Vitro

One of the hallmarks of cancer cell malignancy is the acquisition of cell motility that allows the cancer cell to migrate and invade neighboring tissues and organs. Chemically similar **8b** (pan-HDAC inhibitor) and **9b** (selective HDAC6 inhibitor) were selected to be compared in terms of anti-migratory and anti-invasive properties on MDA-MB-231 and MCF-7 breast cancer cells. Two related in vitro assays were employed: a transwell migration, which mimics cell movement from one culture compartment by crossing through a membrane with 8 μm pores to another; and cell invasion, in which the membrane is coated with a layer of Matrigel that represents a tumor extracellular matrix (ECM), and cells display proteolytic functions to cross through the ECM, among others [45,46]. MDA-MB-231 cells were subjected to migration and invasion in vitro analyses and treated with 5 μM **8b** and 10 μM **9b** for a period of 24 h. To perform the transwell migration and invasion assays with viable cells during the time course of the experiments, these concentrations were selected according to the apoptosis analysis by flow cytometry (Appendix A). As shown in Figure 7A, only **8b** exerted anti-migratory (68% compared to control) and anti-invasive (59% compared to control) properties, as demonstrated by the reduced capacity of MDA-MB-231 cells to move across the wells to the bottom of membranes in both the migration and invasion assay (Figure 7B). Meanwhile, **9b** did not reduce MDA-MB-231 cancer cell migration and invasion compared to control cells in our experimental conditions.

Although HDAC6 has been postulated to play roles in the migration and invasion of cancer cells, mainly by influencing tubulin acetylation and microtubules dynamics [13,47], the use of **9b**, at concentrations sevenfold below the IC_50_, failed to inhibit the TNBC MDA-MB-231 cell migration and invasion. Whether the MDA-MB-231 tubulin cytoskeleton is refractory to HDAC6 inhibition is a matter for future analysis. Nonetheless, an association between estrogen receptor alpha (ERα) and HDAC6 is necessary for the deacetylation of tubulin and increased migration in breast cancer cells [48]. This evidence is consistent with the inhibition by **9b** of ER+ MCF-7 cell line migration (63.9% inhibition of wound closure compared to control) determined by wound healing assay (Appendix A).

#### 3.2.7. Toxicity Assessment In Vivo Using the Zebrafish Model

In order to address whether the most potent newly synthesized HDACi **8b** and **9b** could be applied to humans, we used the zebrafish (*Danio rerio*) as a preclinical animal model, and examined acute and inner organ toxicity upon the applied molecules. Zebrafish have emerged as a universal biotechnological platform for effective and safe drug discovery owing to their genetic, molecular, and immunological similarity to humans, and highly correlated responses to pharmaceuticals, including anti-cancer compounds [49,50,51]. The use of this model system simplifies the path of novel bioactive compounds to clinical trials, and reduces the failure of potential therapeutics at later stages of testing [52,53].

Here, we found that neither of the two novel HDACi isoforms caused a lethal effect at doses up to 100 µM after the 5-day treatment (LC_50_ > 100 µM) (Figure 8A). The data obtained in this assay showed a better toxicology profile of **8b** compared to **9b**. While none of the embryos exposed to 50 µM of **8b** showed signs of teratogenicity, or cardio- or hepatotoxicity (Figure 8B), 32% embryos treated with **9b** were teratogenic (malformed head, jaw, body), had pericardial edema, and a necrotic and small liver. Interestingly, the embryos exposed to 25 and 50 µM of **8b** had reduced circulation in the caudal region, indicating the possible inhibition of angiogenesis by the applied molecule.

#### 3.2.8. Anti-Angiogenic Effects of **8b** and **9b** on Developing Zebrafish

Prompted by the observation of reduced circulation in zebrafish embryos treated with **8b**, we explored the capability of this HDAC inhibitor to inhibit the process of neoangiogenesis in vivo, which is a prerequisite for cancer growth, invasion, and metastasis [54]. While controllable and balanced angiogenesis is essential for normal physiological processes, excessive angiogenesis is important for tumor development and tumor cell metastasis [54]. Accordingly, the inhibition of new blood vessel formation is a proven clinical strategy for treating solid tumors, which, combined with cytostatic treatment, noticeably increases the efficacy of chemotherapy, and provides significantly better survival rates among cancer patients [55].

Here, the suppression of angiogenesis was studied using transgenic zebrafish Tg(fli1:EGFP) embryos, in which the endothelial cells express EGFP, enabling us to directly assess the effect of the applied HDAC inhibitor on vessel development via fluorescence microscopy. Embryos were exposed to **8b** doses ranging from 6.25 to 50 µM (corresponding to non-toxic doses) and imaged for intersegmental vessel (ISV) development after 48 h of treatment. In normally developing embryos, 28–30 ISVs were present. The anti-angiogenic phenotype was defined as the reduced number and/or length of ISVs along the whole body. As shown in Figure 9, **8b** effectively inhibited ISV angiogenesis in a dose-dependent manner. The treated embryos already displayed anti-angiogenic phenotypes at the **8b** dose of 6.25 µM, and ~60% of the embryos were affected in terms of ISV growth (Figure 9B, *p* < 0.05, X^2^ test); on the other hand, the majority of ISVs were inhibited at 25 and 50 µM doses of **8b** (Figure 9C,D). It is important to emphasize that the treatment with the effective doses of **8b** did not elicit any toxic response in the treated embryos, indicating its safety as an anti-angiogenic regime. On the other hand, sunitinib-malate (Suten), a clinical anti-angiogenic drug, provoked life-threatening pericardial edema (Figure 9A), which progressively decreased embryo survival by 120 hpf. Toxicity issues, particularly cardiotoxicity, limit the clinical use of sunitinib at its higher doses and for prolonged periods of time, which restricts its overall anti-angiogenic potential and efficacy as an applied therapy [56].

#### 3.2.9. Compound **8b** Inhibits MDA-MB-231 Breast Tumor Development and Successfully Prevents Tumor Cell Metastasis

The data on the potent inhibition of MDA-MB-231 cell migration and invasion in vitro (Figure 7) justified the evaluation of the anti-metastatic activity of **8b** in vivo. We used the zebrafish MDA-MB-231 xenograft model, and examined the effect of the applied HDACi on tumor development and tumor cell metastasis. Zebrafish xenografts represent a powerful platform for translational research in human carcinomas, demonstrating the crucial hallmarks of cancer biology, such as tumor cell proliferation, dissemination, metastasis, extravasation, and tumor-driven angiogenesis [57]. The use of this model provides differential discrimination on anti-cancer therapy efficacy with single-cell resolution [58]. Accordingly, MDA-MB-231 cells were fluorescently labeled and injected into the yolk of Tg(fli1:EGFP) embryos. At 3 days post treatment (dpt), MDA-MB-231 xenografts were processed for fluorescence microscopy and examined for the effect of **8b** treatment on tumor mass development and cancer cell dissemination and metastasis.

The results obtained in this assay reveal that the treatment with **8b** inhibited both MDA-MB-231 tumor mass development (*p* < 0.001; Figure 10B) and cancer cell dissemination (Figure 10C,D; *p* < 0.001). Compared to the tumor mass developed in the untreated (control) embryos after 3 days post injection, **8b** reduced the mass of breast carcinoma cells by 56.1 ± 5.2% at 3.13 µM dose, while an almost complete reduction was achieved at 12.5 µM (*p* < 0.001 for both doses; Figure 10A,B). Remarkably, no toxic side effects were observed in the xenografts receiving **8b** treatment, which is of a particular relevance since many clinically approved anti-cancer drugs are cardio- and hepatotoxic, which limits their long-term application in chemotherapy. In our analysis of the data for tumor mass inhibition after the 3-day treatment, we determined that the ED_50_ of **8b** (effective drug concentration reducing tumor mass by 50% in relation to the control) was 2.67 µM. Considering that LC_50_ for **8b** was >100 µM, this is evidence that this HDACi possesses a large therapeutic window (determined as the LC_50_/ED_50_ ratio), and clearly indicates that **8b** is a novel, prospective anti-cancer molecule.

In addition to the anti-tumor effect, the newly synthesized HDACi **8b** exhibited potent anti-metastatic activity (Figure 10C,D). Dissemination of MDA-MB-231 cancer cells through the body of xenografts was significantly reduced at the **8b** dose of 3.13 µM. In comparison to the untreated xenografts, the number of embryos exhibiting metastases was reduced by 50% (Figure 10C) and the number of metastasis per embryo was up to 9.6-fold lower (Figure 10D) after 3-day treatment with 3.13 µM **8b**. Moreover, fluorescence microscopy showed that dissemination of MDA-MB-231 cells was completely inhibited at 12.5 µM of **8b** (Figure 10A). It is also important to note that the ISV vasculature of MDA-MB-231 xenografts stayed functional and visible during the entire treatment with **8b**, which indicates that (i) applied treatment has no negative effect on established vasculature, (ii) existing vasculature provides nutrient supply to tumor cells and their dissemination, and (iii) inhibition of tumor cell dissemination was not due to existing vasculature impairment, but due to the anti-metastatic activity of **8b**.

## 4. Conclusions

In this work, 1-benzhydryl piperazine was employed as a novel surface recognition (CAP) group to design potent HDAC inhibitors with anti-metastatic effects in breast cancer. Batches of eight hydroxamic acid derivatives of 1-benzhydryl piperazine were synthesized by varying the nature of the hydrocarbon linker. Two novel selective HDAC6 inhibitors (**6b**, IC_50_ = 0.186 μM and **9b**, IC_50_ = 0.031 μM) were identified, exhibiting more than 10- to 110-fold selectivity over nuclear isoforms HDAC1/3/8. Both potency and selectivity profiles of synthesized selective HDAC6i are comparable with the preclinical drug candidate tubastatin A. The analogues with six and seven carbon atoms in the linker (**7b** and **8b**) were associated with nanomolar potencies, but were non-selective HDACi isoforms. The preliminary SAR analysis supported with structure-based molecular modeling suggested that five carbon atoms (**6b**) is the optimal linker length for achieving selective HDAC6 inhibition among the group of alkyl-hydroxamic acid derivatives. Phenyl-hydroxamic acid derivative (**9b**) demonstrated superior HDAC6 selectivity profile among other synthesized 1-benzhydryl piperazine derivatives.

Evaluation of anti-cancer effects in breast cancer cells (MDA-MB-231) indicated that synthesized HDACi induced apoptosis via the loss of mitochondrial membrane potential. Cellular cytotoxicity of **7b**, **8b**, and **9b** is related to the induction of early apoptosis in the MDA-MB-231 cells and cell cycle arrest in the S phase after 72 h. Compound **8b** effectively decreased the migratory and invasive potential of triple-negative breast cancer cell line (MDA-MB-231), whereas **9b** showed an anti-migratory effect in the wound healing assay toward the MCF-7 cell line. Analysis of cell viability, growth kinetics, and sphere formation in a 3D tumor sphere model of breast cancer singled out **8b** as the most promising anti-cancer compound among the synthesized HDAC inhibitors.

Based on the experiments using zebrafish model, **8b** is disclosed as a novel pan-HDAC inhibitor with superior anti-cancer and anti-metastatic properties, in contrast to selective HDAC6 inhibitor **9b**. Our chemical and biological study of 1-benzhydryl-piperazine-based HDAC inhibitors illustrates how side chain modifications (one carbon atom in aliphatic linkers and the introduction of an aromatic linker) can be exploited to alter the anti-cancer and anti-metastatic properties of HDAC inhibitors.

## Figures and Tables

**Figure 1 pharmaceutics-14-02600-f001:**
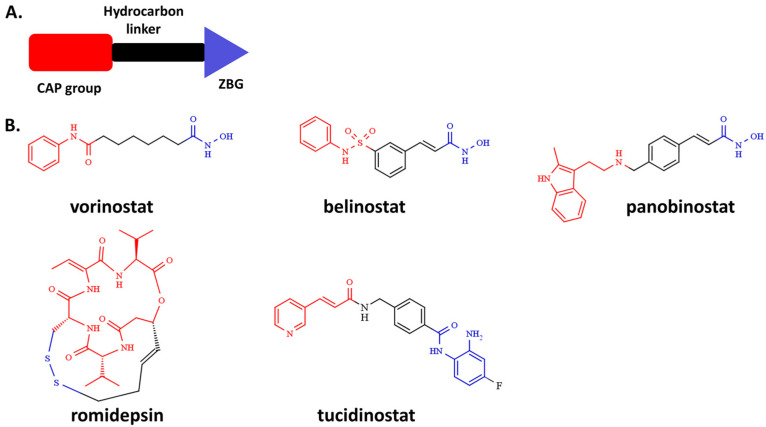
The general pharmacophore model for HDAC inhibitors (**A**) and chemical structures of five FDA-approved HDAC inhibitors (**B**).

**Figure 2 pharmaceutics-14-02600-f002:**
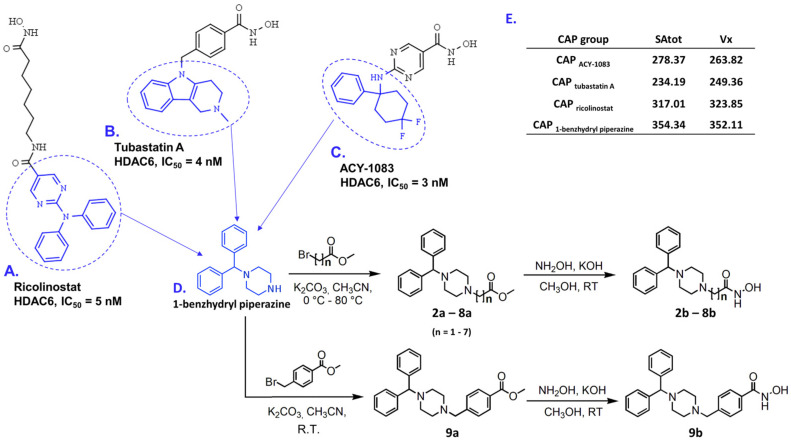
Examples of selective HDAC6 inhibitors: (**A**) ricolinostat, (**B**) tubastatin A, and (**C**) ACY-1083, in which CAP groups are labeled in blue; (**D**) synthetic routes to produce final products **2b**–**9b** starting from 1-benzhydryl piperazine (CAP group) are presented on the bottom right corner, in which n represents the number of carbon atoms in the hydrocarbon linker; (**E**) table with calculated SAtot and Vx descriptors for selected CAP groups and 1-benzhydryl piperazine as a template, presented in the right upper corner.

**Figure 3 pharmaceutics-14-02600-f003:**
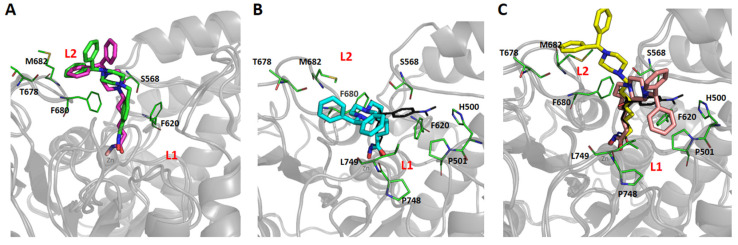
Predicted binding modes of superimposed **6b** (green sticks, (**A**)) and **9b** (magenta sticks, (**A**)), **7b** (cyan sticks, (**B**)), and **8b** (yellow and salmon sticks, (**C**)) in complex with CD2 HDAC6. Interacting amino acid residues from L1 and L2 pockets are represented by green sticks; the co-crystal TSA ligand is presented as black lines.

**Figure 4 pharmaceutics-14-02600-f004:**
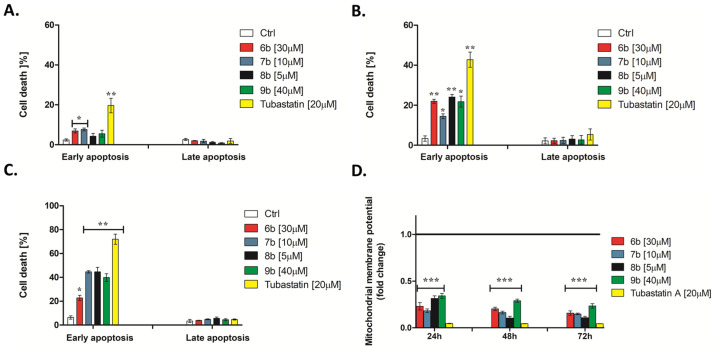
Newly synthesized HDAC inhibitors activate the intrinsic cell death pathway. The proportion of early apoptotic and late apoptotic MDA-MB-231 cells was measured using bivariate Annexin V/7AAD analysis by flow cytometry after the cells underwent treatment with tested compounds at IC_50_ concentration for (**A**) 24 h, (**B**) 48 h, and (**C**) 72 h. (**D**) Dissipation of mitochondrial membrane potential was assessed by flow cytometry using JC-1 dye (compared to the untreated control). Significant differences between treatments were determined by *t*-test: * *p* ≤ 0.05, ** *p* ≤ 0.01, *** *p* ≤ 0.001 (compared to the untreated control).

**Figure 5 pharmaceutics-14-02600-f005:**
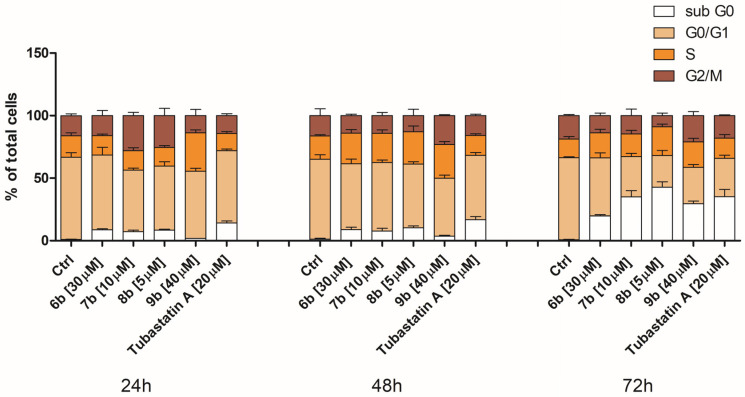
Changes in cell cycle phase distribution of MDA-MB-231 cells upon treatment with compounds **6b**, **7b**, **8b**, **9b**, and tubastatin A. After the 24 h, 48 h, and 72 h continual treatment of MDA-MB-231 cells with investigated compounds at IC_50_, as well as tubastatin A at IC_50_ and 20 μM, cells were stained with propidium iodide and analyzed by flow cytometry. White bar—apoptotic cells with DNA content corresponding to sub-G1 fraction; sandy bar—cells with DNA content corresponding to G0/G1 phases; orange bar—cells with DNA content corresponding to S phase; maroon bar—cells with DNA content corresponding to G2/M phases (compared to the untreated control).

**Figure 6 pharmaceutics-14-02600-f006:**
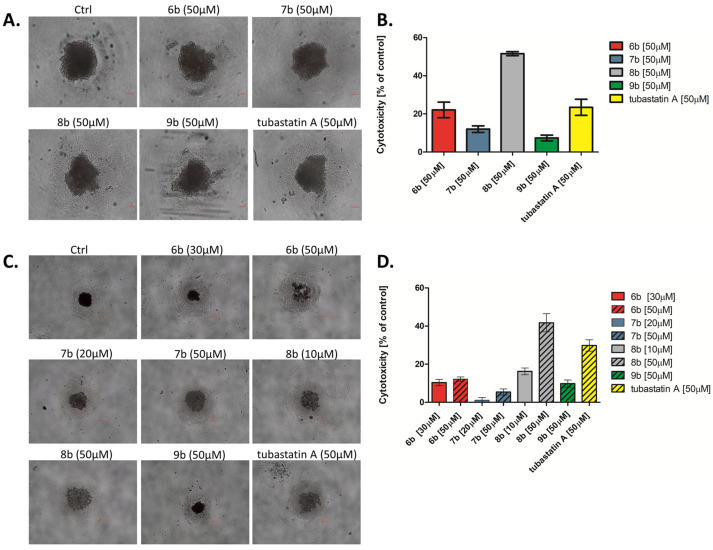
Inhibition of cell survival and growth of MDA-MB-231 tumor spheres treated with synthesized compounds. After 4 d, formed spheres were treated with synthesized compounds and tubastatin A at equimolar concentration (50 μM) for 72 h. The formation and growth of tumor spheres were examined and imaged with an Olympus CKX53, using 4×/0.13 and 10×/0.25 objectives. Scale bar: 100 μm. (**A**) MDA-MB-231 tumor spheres observed under the bright field, and (**B**) the cytotoxicity of compounds toward the MDA-MB-231 tumor spheres was investigated by MTT assay. (**C**) MDA-MB-231 cells were co-cultured with the tested compounds (IC_50_ and 50 µM) in sphere culture conditions for 4 d. The formation and growth of tumor spheres were examined and imaged with an Olympus CKX53, using a 4×/0.4 objective. Scale bar: 200 μm. (**D**) The cytotoxicity of compounds in the co-treatment with MDA-MB-231 tumor spheres was investigated by MTT assay.

**Figure 7 pharmaceutics-14-02600-f007:**
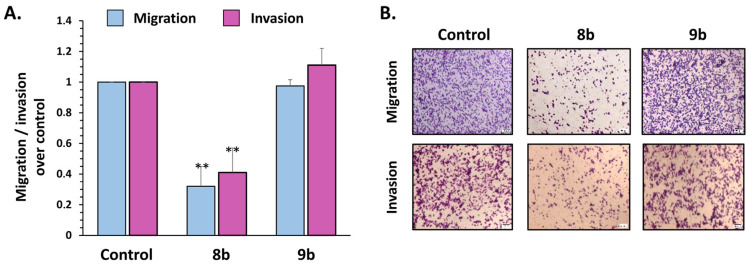
Compound **8b** inhibits migration and invasion of MDA-MB-231 cells (**A**). MDA-MB-231 cells were subjected to bicameral migration (blue) and invasion (purple) assay. Cells were treated with 5 μM of **8b** and 10 μM of **9b** for 24 h. Cells in the bottom of the 8 mm pore membrane were fixed and quantified. Microphotographs show crystal-violet-stained migratory and invasive cells (**B**). Magnification 40×, scale bar = 50 μm. Representative results from three independent experiments are shown. Significant difference between treatments was determined by *t*-test: ** *p* < 0.01.

**Figure 8 pharmaceutics-14-02600-f008:**
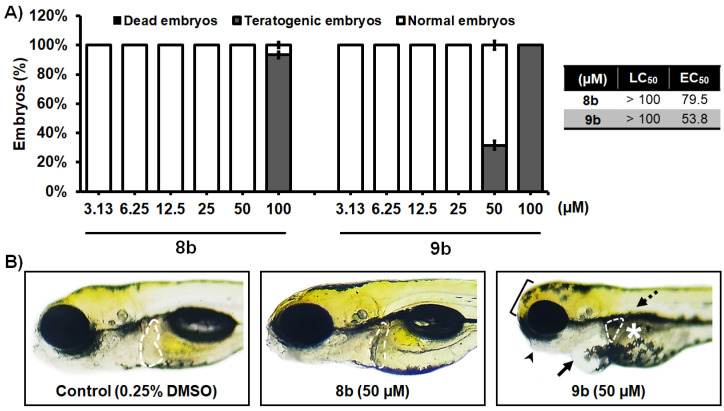
In vivo toxicity assessment of 1-benzhydryl-piperazine-based HDAC inhibitors **8b** and **9b** in the zebrafish (*Danio rerio*) model. (**A**) Acute toxicity was assessed using wild type (AB) embryos exposed to different doses of tested molecules, and expressed as the LC_50_ and EC_50_ doses. Embryos were treated at 6 h post fertilization (hpf), and evaluated for survival, teratogenicity, cardiotoxicity, and hepatotoxicity at 120 hpf (*n* = 60 per a dose). (**B**) Embryos exposed to 50 µM of **8b** were normally developed embryos, without signs of cardiotoxicity, hepatotoxicity, and teratogenicity. On the other hand, treatment with **9b** in some embryos provoked pericardial edema (arrow), weakly resorbed yolk (asterisk), malformed head (bracket) and jaw (arrowhead), no inflated swim bladder (dashed arrow), and decreased and dark liver (outlined). No adverse effect on the liver was detected at 50 µM of **8b**, while reduced liver size and darkening (necrosis) occurred in embryos exposed to 50 µM of **9b**.

**Figure 9 pharmaceutics-14-02600-f009:**
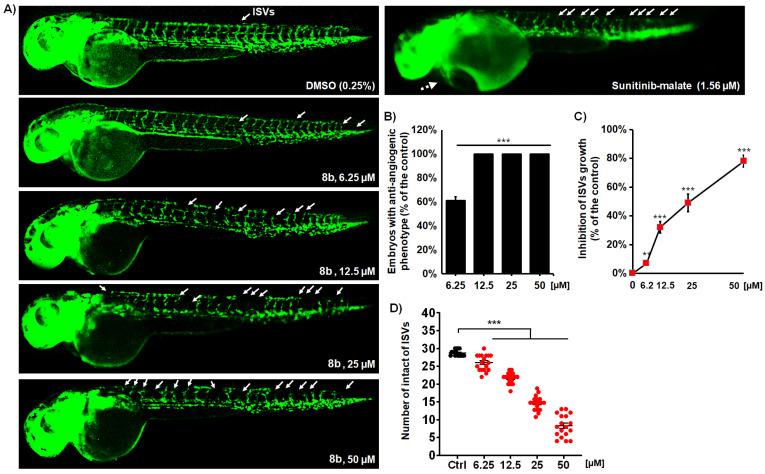
Compound **8b** effectively inhibits angiogenesis in Tg(fli1:EGFP) zebrafish embryos with fluorescently labeled endothelial cells. Embryos were exposed to the four doses of **8b** and assessed the inhibition of ISV vasculature at 48 hpf (**A**). Treatment with **8b** increased the number of embryos with the anti-angiogenic phenotype (**B**), decreased the number of normally developed ISVs (**C**), and reduced the ISV length (**D**) in the dose-dependent manner. Sunitinib-malate (Suten), a clinically approved anti-angiogenic drug, was used as a positive control. While the Suten caused life-threating cardiotoxicity at the effective dose of 1.56 µM, **8b** inhibited angiogenesis without any toxic response in the treated embryos. Representative images of embryos are shown. Data are normalized in relation to the control group (**B**,**C**). Significance in the analyzed parameters between **8b**-treated embryos and control (DMSO-treated) embryos is indicated with asterisks (** *p* < 0.01; *** *p* < 0.001).

**Figure 10 pharmaceutics-14-02600-f010:**
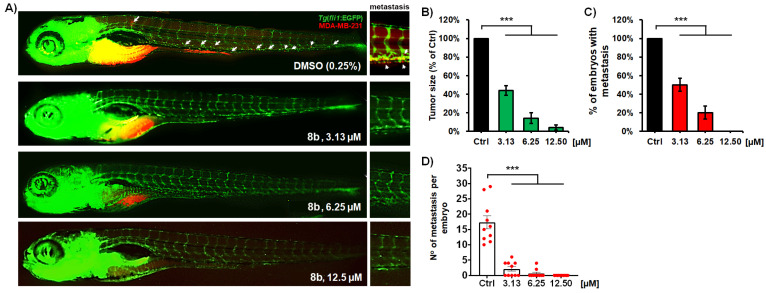
Anti-cancer and anti-metastatic activity of a new HDAC inhibitor **8b** against highly metastatic human breast carcinoma MDA-MB-231 cells in zebrafish xenografts. Tg(fli1:EGFP) xenografts (*n* = 20) were exposed to three different doses of **8b**, and analyzed for tumor progression and metastasis after 3-day treatment. Representative fluorescent microscopy images are shown (**A**); white arrows indicate disseminated cells. The applied treatments markedly reduced the MDA-MB-231 tumor growth (**B**), number of embryos with metastasis (**C**), as well as the number of disseminated cancer cells per xenograft (**D**), as compared to those in the control group (*p* < 0.001, for all hallmarks). Data are normalized in relation to the control group ((**B**–**D**) *** *p* < 0.001).

**Table 1 pharmaceutics-14-02600-t001:** HDAC6 inhibition data for **2b** to **9b**.

	% HDAC6 Inhibition at 5 μM							
Compound	2b	3b	4b	5b	6b	7b	8b	9b
	12.2 ± 1.7	22.2 ± 1.1	3.1 ± 1.7	18.2 ± 1.7	95.6 ± 0.9	97.1 ± 0.8	97.7 ± 0.2	96.9 ± 0.3

Percentage inhibitions are expressed as means ± standard errors of triplicate measurements.

**Table 2 pharmaceutics-14-02600-t002:** Enzymatic in vitro profiles of novel hydroxamic acid 1-benzhydryl piperazine derivatives against HDAC1, HDAC3, HDAC6, and HDAC8 isoenzymes.

Compound	HDAC Inhibition, IC_50_ ± SD (μM)				Selectivity Ratio		
	HDAC1	HDAC3	HDAC6	HDAC8	HDAC1/6	HDAC3/6	HDAC8/6
**6b**	4.730 ± 0.670	1.860 ± 0.090	0.186 ± 0.005	2.440 ± 0.510	25.4	10	13.1
**7b**	0.619 ± 0.023	0.267 ± 0.019	0.096 ± 0.008	0.345 ± 0.017	6.4	2.8	3.6
**8b**	0.408 ± 0.022	0.207 ± 0.014	0.124 ± 0.013	0.245 ± 0.002	3.3	1.7	2
**9b**	1.474 ± 0.172	3.467 ± 0.121	0.031 ± 0.004	0.708 ± 0.045	47.5	111.8	23
Valproic acid ^#^	660	>1000	>1000	103	-	-	-
Sodium butyrate ^#^	8.3	4.8	>1000	10.4	-	-	-
Trichostatin A ^#^	0.0013	0.0015	0.0036	0.400	0.3	0.4	111.1

Values are expressed as the mean ± SD of three independent experiments. ^#^ Profiles of reference HDACi [27].

**Table 3 pharmaceutics-14-02600-t003:** Cytotoxic activities toward MDA-MB-231 and MCF-7 cell lines.

Compound	IC_50_ ± SD (μM)	
	MDA-MB-231	MCF-7
**6b**	33.40 ± 2.79	84.05 ± 5.2
**7b**	10.55 ± 1.95	>100
**8b**	5.42 ± 0.77	39.10 ± 2.7
**9b**	38.21 ± 3.01	99.50 ± 0.7
**tubastatin A**	20.83 ± 2.84	93.31 ± 9.4

Values are expressed as the mean ± SD of three independent experiments.

## Data Availability

The raw data supporting the conclusions of this manuscript will be made available by the authors, without undue reservation, to any qualified researcher.

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
