# Peer review of "Discovery of 1-Benzhydryl-Piperazine-Based HDAC Inhibitors with Anti-Breast Cancer Activity: Synthesis, Molecular Modeling, In Vitro and In Vivo Biological Evaluation"

_pharmaceutics, 2022, doi:10.3390/pharmaceutics14122600_

Round 1
Reviewer 1 Report
The manuscript entitled “Discovery of 1-benzhydryl piperazine-based HDAC inhibitors with anti-cancer and anti-metastatic properties against human breast cancer: synthesis, molecular modeling, in vitro and in vivo biological evaluation” reports the design, synthesis and characterization of 1-benzhydryl piperazine derivatives as potent HDAC inhibitors. All compounds were screened for their HDAC inhibition activities, and compounds 6b, 7b, 8b and 9b were evaluated for their cytotoxic effects against MDA-MB-231 and MCF-7 breast cancer cell-lines. Authors found that the compound 8b exhibited remarkable activities as the HDAC inhibitor with very potent anti-tumor, anti-metastatic and anti-angiogenic effects in zebrafish MDA-MB-231 xenograft models at low micromolar concentrations.
Overall the manuscript is rich and interesting; and the paper structure is well-knit and suitable for publication in the journal, after minor revisions. The comments are listed as the following points:
1- In abstract, line 29, “9b” should be in bold.
2- The IC50 was expressed in micromolar in Table 2, but in the text the IC50 was expressed in nanomolar. The units should be homogenized.
3- In Figure 1, correct the structure of “tucidinostat”. The fluoro group should be in position-4 instead of position-5.
4- In Figure 1, correct the structure of “Romidepsin”. Show the asymmetric carbons (chiral carbons).
5- Title of Table 1, line 347, “2b and 9b” should be in bold.
6- Lines 427 and 429 should be written in the same font as the text.
7- Lines 677 and 680, “8b” should be in bold.
8- Conclusion should be shortened.
9- Line 691, Add titles for Figures S1-S8 and Tables S1 and S2.
10- Besides, authors should provide HRMS images (in Supplementary data) for purity verification.
Author Response
Dear Editor and Reviewers,
Thank you for your letter and the reviewers’ comments concerning our manuscript entitled “Discovery of 1-benzhydryl piperazine-based HDAC inhibitors with anti-cancer and anti-metastatic properties against human breast cancer: synthesis, molecular modeling, in vitro and in vivo biological evaluation” (Manuscript number: pharmaceutics-2038114). These comments are all valuable and very helpful for revising and improving our paper. We have studied comments carefully and have made correction which we hope meet with approval. We used track changes mode in the manuscript to label minor corrections suggested by reviewers. The responds to the reviewer’s comments are given below:
Responds to the reviewer’s comments:
Reviewer 1:
The manuscript entitled “Discovery of 1-benzhydryl piperazine-based HDAC inhibitors with anti-cancer and anti-metastatic properties against human breast cancer: synthesis, molecular modeling, in vitro and in vivo biological evaluation” reports the design, synthesis and characterization of 1-benzhydryl piperazine derivatives as potent HDAC inhibitors. All compounds were screened for their HDAC inhibition activities, and compounds 6b, 7b, 8b and 9b were evaluated for their cytotoxic effects against MDA-MB-231 and MCF-7 breast cancer cell-lines. Authors found that the compound 8b exhibited remarkable activities as the HDAC inhibitor with very potent anti-tumor, anti-metastatic and anti-angiogenic effects in zebrafish MDA-MB-231 xenograft models at low micromolar concentrations.
Overall the manuscript is rich and interesting; and the paper structure is well-knit and suitable for publication in the journal, after minor revisions. The comments are listed as the following points:
- In abstract, line 29, “9b” should be in bold.
Author response: We have changed this in the manuscript.
- The IC50 was expressed in micromolar in Table 2, but in the text the IC50 was expressed in nanomolar. The units should be homogenized.
Author response: We have changed the nM concentration units to the μM concentrations and labelled all corrections in track changes.
- In Figure 1, correct the structure of “tucidinostat”. The fluoro group should be in position-4 instead of position-5.
Author response: We have corrected the structure of tucidinostat, the fluoro- group is in position 4.
- In Figure 1, correct the structure of “Romidepsin”. Show the asymmetric carbons (chiral carbons).
Author response: The structure of Romidepsin is changed and all the asymmetric carbon atoms are clearly labelled on a new Figure 1.
- Title of Table 1, line 347, “2b and 9b” should be in bold.
Author response: The codes for the compounds 2b and 9b are now in bold.
- Lines 427 and 429 should be written in the same font as the text.
Author response: The title of the Table 3 (line 427) and its description bellow (line 429) are now written in the same font (Palatino Linotype).
- Lines 677 and 680, “8b” should be in bold.
Author response: The compound “8b” is now written in bold in the lines 677 and 680.
- Conclusion should be shortened.
Author response: With a respect to the reviewer’s comment, we found that the third paragraph was the most appropriate to be shortened while retaining the same summary of findings within the paper. Initially, from 358 words, novel conclusion has 322 words, that is almost 10% shortened.
- Line 691, Add titles for Figures S1-S8 and Tables S1 and S2.
Author response: The titles for all the Supporting Figures, Tables and Note 1 are added to the Supplementary Materials section.
- Besides, authors should provide HRMS images (in Supplementary data) for purity verification.
Author response: The HRMS images for the active compounds are given in the Supplementary data, titled as LC-ESI (+) HRMS (ToF) mass spectra for compounds 6b – 9b in a range (m/z 100–1000), now in a separate section, after 1H and 13CNMR spectra.

Reviewer 2 Report
In this study, Ruzic et al., discover new 1-benzhydryl piperazine-based selective HDAC inhibitors with anti-cancer and anti-metastatic properties in breast cancer. Authors have designed new HDAC inhibitors using a novel approach and show promising effects on breast cancer cells. Below are some of the minor comments:
1. How does the expression of HDAC6 changes upon the treatment of these inhibitors’ both at RNA and protein levels?
2. What is the efficacy of these inhibitors in other cancers expressing HDAC6?
3. What is the specificity of HDAC6 inhibitors on other HDACs?
4. Authors should provide the expression levels of apoptotic markers at deciphered IC50 values and at different time points.
5. How does these inhibitors impact chromatin remodeling and expression of target genes?
Author Response
Dear Editor and Reviewers,
Thank you for your letter and the reviewers’ comments concerning our manuscript entitled “Discovery of 1-benzhydryl piperazine-based HDAC inhibitors with anti-cancer and anti-metastatic properties against human breast cancer: synthesis, molecular modeling, in vitro and in vivo biological evaluation” (Manuscript number: pharmaceutics-2038114). These comments are all valuable and very helpful for revising and improving our paper. We have studied comments carefully and have made correction which we hope meet with approval. We used track changes mode in the manuscript to label minor corrections suggested by reviewers. The responds to the reviewer’s comments are given below:
Reviewer 2:
In this study, Ruzic et al., discover new 1-benzhydryl piperazine-based selective HDAC inhibitors with anti-cancer and anti-metastatic properties in breast cancer. Authors have designed new HDAC inhibitors using a novel approach and show promising effects on breast cancer cells. Below are some of the minor comments:
- How does the expression of HDAC6 changes upon the treatment of these inhibitors’ both at RNA and protein levels?
Author response: After thorough search of the relevant literature, we have not found information regarding the effects of HDAC6 inhibitors on RNA and protein expression levels of HDAC6. The quite intriguing fact is that HDAC6 is located mostly in the cytoplasm, it deacetylates cytoplasmic proteins including tubulin, cortactin and Hsp90. HDAC6 inhibition leads to hyperacetylation of tubulin and stabilization of microtubules. As Hsp90 regulates the transcription factors network, hyperacetylation of Hsp90 has “indirect” effects on the regulation of gene expression (http://dx.doi.org/10.1016/j.bcp.2012.06.014). To conclude, HDAC6 mainly regulates the Hsp90-HSF1 complex, thus promoting the HSF1 entry into the nucleus and activating tumor-suppressor genes transcription (doi.org/10.1186/s10020-021-00375-3).
- What is the efficacy of these inhibitors in other cancers expressing HDAC6?
Author response: Many thanks for this interesting question. As the HDAC enzymes are also implicated in other malignancies, after encouraging results on breast cancer, we were interested to examine the effects of novel active HDAC inhibitors (6b – 9b) on pancreatic cancer cell lines. Accordingly, the reviewer is kindly referred to our novel preprint paper that could be found on a link - https://www.biorxiv.org/content/10.1101/2022.10.10.511584v1 .
- What is the specificity of HDAC6 inhibitors on other HDACs?
Author response: To examine the selectivity profiles on HDAC6 inhibitors, we performed in vitro enzymatic assays on Class I HDAC representatives (HDAC1, HDAC3 and HDAC8), as they are homologous to HDAC class IIb (HDAC6 and HDAC10). It could be an interesting point to assay the compounds on HDAC10 isoform in future studies, but the question is whether HDAC10 inhibition is relevant for breast cancer therapy (please see Table S1 in Supporting Information). Class IIa HDACs (HDAC4, HDAC5, HDAC7 and HDAC9) are more potently targeted by non-hydroxamate inhibitors due to the catalytic pocket Tyr-His mutation (https://doi.org/10.3390/molecules26175151), so we would expect that our compounds are not potent class IIa HDACs inhibitors.
- Authors should provide the expression levels of apoptotic markers at deciphered IC50 values and at different time points.
Author response: Many thanks for this question. Since we observed significant induction of early apoptotic cell death and dissipation of mitochondrial membrane potential in time-dependent manner that is followed with consistently low percentage of cells in the late apoptotic stage, it may be concluded that markers involved in the intrinsic apoptotic pathway are changed (caspase-8/9, Bax, Bcl-2, Apaf-1). This is an important task that will be examined in our future studies from mechanistic point of view. For this medicinal chemistry project, we aimed to demonstrate that in silico drug design may lead to identification of novel HDAC inhibitors, active in cell-based assays and in vivo model.
- How does these inhibitors impact chromatin remodeling and expression of target genes?
Author response: Many thanks for this question. As HDAC inhibition is found to increase the level of acetyllysine residues on histone tails, this is associated with loss of positive charge on histones, which at the end cause relaxation of chromatin structure. Oppositely, aberrant histone deacetylation in numerous cancers causes heterochromatin condensation that supresses transcription. Finally, it is known that HDAC inhibitors affect expression of numerous tumor-suppressor genes (p53, c-Myc, HIF-1 α, estrogen receptor, β-catenin, STAT3 and many more) (https://doi.org/10.1016/j.bioorg.2019.103184), therby inducing open chromatin conformation at tumor suppressor gene loci, cell-cycle arrest, chemo-sensitization, apoptosis and upregulation of tumor suppressors (https://doi.org/10.1016/j.lfs.2021.119504).

Reviewer 3 Report
In the present manuscript entitled “Discovery of 1-benzhydryl piperazine-based HDAC inhibitors with anti-cancer and anti-metastatic properties against human breast cancer: synthesis, molecular modeling, in vitro and in vivo biological evaluation” by Ruzic et al designed and synthesized a series of 1-benzhydryl piperazine-based analogs and evaluated their HDAC6 inhibition and in-vitro enzymatic activity against HDAC panel. The authors found that two analogs inhibited HDAC6 (6b and 9b) at nanomolar concentrations. They have studied in silico molecular modelling and anti-cancer activity in MDA-MB-231 and MCF-7 cellular system. They also well characterized all synthesized compounds by using standard spectroscopic techniques such as 1HNMR, 13CNMR and HR-MS. This is an interesting paper, well drawn manuscript, well-written and organized and supported by a comprehensive and up to date literature. I would recommend the manuscript for publication in “Pharmaceutics” after minor revisions.
1. There is a need for more discussion about 8b (cellular permeability), as it shows 6-7-fold greater potency in cellular system than 6b and 9b analogs.
2. It would be more addition to this paper if authors study in vitro PK profiles such as metabolic stability, aqueous solubility, and pH stability.
Author Response
Dear Editor and Reviewers,
Thank you for your letter and the reviewers’ comments concerning our manuscript entitled “Discovery of 1-benzhydryl piperazine-based HDAC inhibitors with anti-cancer and anti-metastatic properties against human breast cancer: synthesis, molecular modeling, in vitro and in vivo biological evaluation” (Manuscript number: pharmaceutics-2038114). These comments are all valuable and very helpful for revising and improving our paper. We have studied comments carefully and have made correction which we hope meet with approval. We used track changes mode in the manuscript to label minor corrections suggested by reviewers. The responds to the reviewer’s comments are given below:
Reviewer 3:
In the present manuscript entitled “Discovery of 1-benzhydryl piperazine-based HDAC inhibitors with anti-cancer and anti-metastatic properties against human breast cancer: synthesis, molecular modeling, in vitro and in vivo biological evaluation” by Ruzic et al designed and synthesized a series of 1-benzhydryl piperazine-based analogs and evaluated their HDAC6 inhibition and in-vitro enzymatic activity against HDAC panel. The authors found that two analogs inhibited HDAC6 (6b and 9b) at nanomolar concentrations. They have studied in silico molecular modelling and anti-cancer activity in MDA-MB-231 and MCF-7 cellular system. They also well characterized all synthesized compounds by using standard spectroscopic techniques such as 1HNMR, 13CNMR and HR-MS. This is an interesting paper, well drawn manuscript, well-to date literature. I would recommend the manuscript for publication in “Pharmaceutics” after minor revisions.
- There is a need for more discussion about 8b (cellular permeability), as it shows 6-7-fold greater potency in cellular system than 6band 9b
Author response: Many thanks for this interesting point of view and question. We definitively agree that the cellular permeability should be thoroughly examined to explain differences in effects observed in cell-based assays. We anticipate that compounds’ lipophilicity play an important role, thus we predicted their logP and logD values in ADMET Predictor (the results are given bellow):
|
Structure |
Identifier |
S+logP |
S+logD |
S+pH (mg/mL) |
CYP_Risk and CYP Code |
|
6b |
2.582 |
1.927 |
1,41 |
1,589 (3A4), position C4 and C4’ |
|
|
7b |
2.978 |
2.281 |
1,157 |
2,033 (2D6; 3A4), position C4 and C4’ |
|
|
8b |
3.378 |
2.66 |
0,9 |
2,113 (2D6; 3A4), position C4 and C4’ |
|
|
9b |
2.876 |
2.724 |
0,051 |
1,229 (3A4), position C4 and C4’ |
S+logP is octanol-water partition coefficient, log P;
S+logD is octanol-water distribution coefficient, log D; calculated from pKa and S+logP.
S+pH (mg/mL, native pH calculated from ionization constants and native water solubility; requires pKa)
CYP_Risk and CYP_Code (ADMET Risk and ADMET Code for metabolic liability)
As may be observed from the Table above, compound 8b is predicted with the highest lipophilicity (logP=3.378) among synthesized analogues. This may contribute to its highest permeability inside the MDA-MB-231 cells. We are also aware that other mechanisms contribute to different permeation of 6b-9b inside MDA-MB-231 cells, therefore in this study we avoided to include single in silico pharmacokinetic predictions without experimental validation.
- It would be more addition to this paper if authors study in vitro PK profiles such as metabolic stability, aqueous solubility, and pH stability.
Author response: We kindly acknowledge for this remark. Similarly to the previous question, we calculated (predicted) aqueous solubility (S+pH), general metabolic stability (CYP_Risk) in ADMET software. The results are given in the table above. It is important to underline that all the compounds were soluble in cell medium during cell-based assays that is in line with in silico solubility predictions given above. Moreover, it is predicted that compound 8b has the highest probability to be metabolized by CYP enzymes (isoforms 2D6 and 3A4). For all the compounds 6b-9b, it is expected that phenyl rings within 1-benzhydryl piperazine will be oxidized to phenols in positions 4 and 4’.
For this medicinal chemistry project, we aimed to demonstrate that in silico drug design may lead to identification of novel HDAC inhibitors, active in cell-based assays and under in vivo experimental settings. So, our next step will include lead optimization of 8b and 9b inhibitors with careful assessment of DMPK properties.
